# Comparison of Hydroxypropylcellulose and Hot-Melt Extrudable Hypromellose in Twin-Screw Melt Granulation of Metformin Hydrochloride: Effect of Rheological Properties of Polymer on Melt Granulation and Granule Properties

**Amol Batra [1], Fengyuan Yang [1,\*], Michael Kogan [2], Anthony Sosnowik [2], Courtney Usher [2], Eugene W. Oldham [2], Ningyi Chen [1], Kamaru Lawal [1], Yunxia Bi [1] and Thomas Dürig [1]**

[1] Life Sciences R&D and Innovation, Ashland Specialty Ingredients, 500 Hercules Road, Wilmington, DE 19808, USA; amol_batra@ashland.com (A.B.); Ningyi_Chen@ashland.com (N.C.); Kamaru_Lawal@ashland.com (K.L.); Yunxia_Bi@ashland.com (Y.B.); Thomas_Durig@ashland.com (T.D.)

[2] Measurement Science, Ashland Specialty Ingredients, 500 Hercules Road, Wilmington, DE 19808, USA; Michael_Kogan@ashland.com (M.K.); Anthony_Sosnowik@ashland.com (A.S.); Courtney_Usher@ashland.com (C.U.); Eugene_Oldham@ashland.com (E.W.O.)

\* Correspondence: fyyang@ashland.com; Tel.: +1-302-995-3664

**Abstract:** High-molecular-weight hypromellose (HPMC) and hydroxypropyl cellulose (HPC) are widely known, extended-release polymers. Conventional high-molecular-weight HPMCs are preferred in extended-release applications but not widely used in twin-screw melt granulation due to processability difficulties at low operating temperatures and potential drug degradation if high processing temperatures are used. Conversely, high-molecular-weight grade HPC (Klucel®) can be used in melt granulation processes. The purpose of this study was to evaluate the processability and dissolution behavior of HPC GXF ((Klucel® GXF) and a recently introduced type of hot-melt extrudable HPMC (Affinisol®) in extended-release metformin hydrochloride formulations using twin-screw melt granulation. Powder blends were prepared with 75% *w/w* metformin HCl and 25% *w/w* polymeric binder. Blends were granulated at processing temperatures of 160, 140, 120 and 100 °C. HPMC HME 4M (Affinisol® 4M) provided a fine powder, indicating minimum granulation at processing temperatures lower than 160 °C, and the tablets obtained with these granules capped during tableting. In contrast, acceptable tablets could be obtained with HPC GXF at all processing temperatures. Rheological studies including capillary rheometry to measure steady shear rate viscosity, and rotational rheometry to obtain time and temperature superposition data, showed that HPC GXF had a greater thermoplasticity than HPMC HME 4M, which made granulation possible with HPC GXF at low temperatures. Tablets compressed with granules obtained at 160 °C with both binders showed comparable dissolution profiles. High-molecular-weight HPC GXF provided a better processability at low temperatures and adequate tablet strength for the melt granulation of metformin HCl.

**Keywords:** twin-screw melt granulation; tablet; polymer; binder; rheology; melt extrusion; hydroxypropyl cellulose; hydroxypropyl methylcellulose

## 1. Introduction

In pharmaceutical solid dosage form development, granulation is one of the most important unit operation. Granulation in simple terms is particle enlargement by the agglomeration of a drug and excipient in a formulation. Granulation improves powder flow, content uniformity and tablet properties. Granulation, traditionally, is carried out by wet granulation and roller compaction. These processes are batch operations which consist of a series of unit operations [1]. Due to the multiple unit operations involved in manufacture, the overall process is costly and time consuming. Gradually, over the last

decade, the pharmaceutical industry has moved towards continuous manufacturing to produce the final product from raw material in a single step [2]. Continuous manufacturing, therefore, is more economically beneficial than batch manufacturing [3]. In this process of exploration into alternative granulation methods, melt granulation using a twin-screw extruder has emerged as one of the most excellent methods for continuous granulation [4,5].

In melt granulation, a polymeric or waxy material is used to agglomerate fine particles. The polymeric or waxy binder melts at a high temperature to agglomerate drug and excipient particles. Over the last decade, polymeric binders were proven to provide better control and superior granule properties over waxy binders [6–10]. A series of recently launched products in the market use polymeric binders in the process of melt granulation. Eucreas® or Glavumet® (Novartis), a combination product of metformin hydrochloride (metformin HCl) and vildagliptin [10,11], an aliskiren-valsartan fixed dose combination bilayer tablet [12,13], and an imatinib mesylate modified release tablet [14] are manufactured using twin-screw melt granulation.

Polymeric materials are used for many pharmaceutical formulations and drug-delivery applications. The use of these polymeric materials in pharmaceutical application requires a thorough understanding of their structure and properties. The success of a pharmaceutical process depends on the performance of these polymeric materials, which in turn depends on their chemistry. Especially in the case of melt extrusion, a polymer's physiochemical properties, such as chemical composition, molecular weight and molar substitution may have an effect on their thermomechanical properties [15]. A series of reports in the literature show that cellulosic- [16], pyrollidone- [17] and acrylate [18]-based polymers show different rheological behavior based on their chemistry and molecular weight. Due to thermomechanical and rheological properties, it is difficult to extrude some polymeric materials using a twin-screw extruder. Especially when it comes to melt extrusion, polymeric materials such as high-molecular-weight grades of hydroxypropyl methylcellulose (HPMC) and polyvinylpyrrolidone (PVP) may sometimes be used in melt extrusion but have processing difficulties due to their thermomechanical properties. These grades of HPMC and PVP may require the addition of plasticizer to lower the processing temperature for extrusion [19,20]. In contrast, high-molecular-weight HPC does not show any processing difficulties and can be extruded without the addition of plasticizer [21]. Polymers with low glass transition temperatures and a comparatively easier processability in melt extrusion, such as HPC [6,22], hypromellose acetate succinate (HPMCAS) [23,24] and Poly (vinylpyrrolidone-co-vinyl acetate) (copovidone) [25,26] are preferred for the manufacture of amorphous solid dispersions of poorly water-soluble drugs.

The preparation of solid dispersion by melt extrusion is quite different from melt granulation using a twin-screw extruder. In general, solid dispersions utilize more than 60% of a polymer; the active pharmaceutical ingredient (API) is miscibilized in the polymer at high temperature, and the molten material is extruded from the twin-screw extruder as strands that solidify at room temperature. On the other hand, in the case of melt granulation, only about 10–30% polymer is mixed with the API, and the mixture is extruded through a melt extruder at a temperature above the glass transition temperature (Tg) of the polymer but below the melting temperature of the API. Therefore, when using a polymeric binder for melt granulation, it should have an optimum low viscosity to granulate/agglomerate drug particles, even when used at lower concentrations of 10–30% [2]. The impact of a polymeric binder's rheological properties on the final product is even more pronounced in the case of melt granulation. For this reason, high-molecular-weight HPMC or PVP, therefore, are not generally selected for use in melt granulation, and HPC is usually preferred in granulation for processing at low temperatures and with low binder concentration in the formulation [27].

Conventional high-molecular-weight HPMCs are widely accepted to be excellent polymers for extended-release formulations [28]. HPMC is used in different processes such as the direct compaction of tablets, roller compaction, spray drying, etc. The compaction properties of HPMC makes it possible to compress easily into tablets. Conventional HPMCs

are marketed in different molecular weights ranging from HPMC K15M to K100M. These molecular weight grades have different viscosities in water which enables them to provide the desired drug release profiles. Despite the advantages provided by HPMC in tableting and controlling drug release, its use in twin-screw melt granulation is limited. Because of the high glass transition temperatures of high-molecular-weight HPMCs, and thus the high processing temperatures required for operating, it is extremely difficult to extrude them at low temperatures to prevent potential drug and polymer degradation. Due to this, recently, a new grade of HPMC, Affinisol® (HPMC HME), was introduced by IFF (Wilmington, DE, USA) which has a lower glass transition temperature compared to conventional high-molecular-weight HPMCs [29]. HPMC HME, due to its lower glass transition temperature and the ability to be extruded at a lower temperature, provides an advantage in melt extrusion [30]. However, the processing temperature must be greater than the glass transition temperature of the polymer [30,31].

The objective of this study, therefore, was to evaluate the processability and dissolution behavior of comparable molecular weight grades of HPC and a recently introduced type of hot melt extrudable HPMC in extended-release formulations using twin-screw melt granulation.

## 2. Materials and Methods

Metformin HCl was selected as a model of highly water-soluble drug. Metformin HCl was purchased from Harman Finochem Ltd. (Maharashtra, India).

Affinisol® HPMC HME 4M (HPMC HME 4M) is manufactured and marketed by IFF (Wilmington, DE, USA). HPMC HME polymers are pure hydroxypropyl methylcellulose without any additives present, and they differ among themselves and from other HPMC polymers, such as Methocel™ K100LV, in the level of chemical substitution [29,30]. Three molecular weight grades of Affinisol® HPMC HME are available—HPMC HME 15LV, HPMC HME 100LV, and HPMC HME 4M—with approximate molecular weights of 90, 180 and 550 kDa [29], respectively. Since metformin HCl is highly water-soluble, HPMC HME 4M, with the highest molecular weight of approximately 550 kDa, was selected for experimentation.

Klucel® GXF HPC (HPC GXF) was obtained from Ashland Specialty Ingredients (Wilmington, DE, USA). HPC GXF with a molecular weight of 370 kDa was selected. Due to the manufacturability, this grade has the most comparable molecular weight to HPMC HME 4M.

Table 1 lists the chemical name, glass transition temperature and onset of thermal degradation for HPC GXF and HPMC HME 4M.

**Table 1.** Polymers used for melt granulation of metformin hydrochloride.

| Chemical Name | Trade Name (Manufacturer) | $T_g$ (°C) | Degradation Temperature (°C) |
|---|---|---|---|
| Hydroxypropyl cellulose (MW~370 kDa) | Klucel® GXF | 84 [c] | 227 [a] |
| Hydroxypropyl methylcellulose (MW~552 kDa) | Affinisol® 4M | 115 [b] | 220 [b] |

MW, molecular weight; kDa, Kilodalton;]; [a] refer to [16]; [b] refer to Ref. [31]; [c] refer to Ref. [32].

### 2.1. Powder Blending

Metformin HCl is a free-flowing powder and is not hygroscopic by nature. Due to its high water solubility, however, it may form solid bridges between particles and agglomerate to form hard lumps [33]. It is essential to mill or sieve metformin HCl to break these powder lumps before blending. Batra et al. [10] suggested using a high shear mixer for the blending of metformin HCl and polymer. Powder blends were prepared with 75% *w/w* metformin HCl and 25% *w/w* polymeric binder. Therefore, metformin HCl and

polymer were blended in a high shear mixer with the main blade speed and chopper speed of 1250 and 800, respectively.

### 2.2. Twin-Screw Melt Granulation

An 18 mm, Leistritz ZSE 18 HP corotating twin-screw melt extruder (Leistritz Corporation, Nuremberg, Germany) was used for melt granulation. A gravimetric feeder (Coperion K-Tron, Pitman, NJ, USA) was used to introduce powder into the melt extruder at a controlled feed rate. The twin-screw extruder barrel (720 mm in length) was divided into 8 independent heating zones (Zone 1 to Zone 8), each 90 mm in length and capable of maintaining a specific set temperature. The die at the end of barrel was removed to obtain granules. The powder blend (feed) was introduced into Zone 1. In all cases throughout this study, the temperature in Zone 1 was set at a temperature of 20 °C. The maximum temperature variation that was observed for Zone 1 due to conductive heating from Zone 2 was $25 \pm 5$ °C. A temperature profile detailing the heating or cooling temperatures of materials as they travel from Zone 1 (feeding zone) to Zone 8 is described later in this paper. The highest temperature attained in a specific temperature zone is mentioned as the processing temperature for granulation.

### 2.3. Differential Scanning Calorimetry

Thermal analysis of melt-granulated samples was performed using a DSC Q200 (TA instruments, New Castle, DE, USA). A constant nitrogen purge of 20 mL/min was maintained during analysis. Around 6–7 mg of the melt-granulated sample was weighed and hermetically sealed in a pan. Differential scanning calorimetry (DSC) ramp up was performed from 25 °C to 220 °C at a heating rate of 10 °C/min.

### 2.4. Milling

The agglomerates obtained after melt granulation were milled using a 197 overdriven comill (Quadro Engineering Corporation, Waterloo, ON, Canada) and passed through a 1575 μm screen. A round-edged impeller rotating at a tip speed of 2.4 m/s was used. Powder was manually charged to the mill at approximately 1 to 10 kg/h. The conical design of co-mill and centrifugal forces propel the mixed particles outward and up toward the impeller tip and screen. As the agglomerates/particles become trapped between the screen and impeller edge, significant shear stresses are imparted. After shearing, some of these particles pass through the screen open area and the remaining contained particles are displaced back into the center mixing zone. Eventually, all of the particles pass through the screen until the entire charge volume is emptied.

### 2.5. Granule Size Analysis

To study the effect of milling on granule size, different granule size fractions were collected by sieve analysis. US sieves # 16, 18, 20, 30, 45, and 120 with mesh apertures of 1180, 1000, 850, 600, 355, and 125 μm, respectively, were used for granule size analysis. A total batch size of 100 g was used for sieve analysis.

### 2.6. Tablet Compaction

Granules obtained after twin-screw melt granulation were blended with 1% $w/w$ of magnesium stearate for 2 min using a turbula mixer (Willy A. Bachofen AG Maschinenfabrik, Basel, Switzerland). Granules were compacted at two different compaction forces of 15 and 30 kN to obtain 1000 mg tablets (containing 750 mg of metformin HCl) using a STYLONE® compaction simulator (Medelpharm Instruments, Beynost, France). The compaction simulator was used to stimulate tableting conditions used during high-speed tableting in commercial manufacturing. The STYLONE® compaction simulator was set to simulate Manesty® Betapress operating at 33 rpm (31277 tablets/h) using 0.374 × 0.748 in. oval punch and die (Natoli Engineering, Saint Charles, MO, USA).

The tablet hardness was measured using diametrical hardness testing unit (Key International Inc., Englishtown, NJ, USA). The tablet hardness obtained in kilogram-force (kgf) was converted to Newtons (N). An increase in tablet hardness was expected after melt granulation of metformin HCl with polymeric binders.

### 2.7. Dissolution

Dissolution testing was performed in pH 6.8 phosphate buffer (1000 mL) using USP apparatus 1 at 37 °C and 100 rpm for 24 h [34]. Absorbance of analyte was measured at 230 nm using an Agilent 1100 with a UV spectrophotometer to calculate drug concentration.

### 2.8. Scanning Electron Microscopy (SEM)

Tablets that had undergone dissolution for one hour were flash frozen in liquid nitrogen, then freeze-dried for 24 h before being fractured with a razor blade to expose the internal surface. Samples were mounted on aluminum sample stubs, coated with a thin layer of Au/Pd to make the sample surface conductive and then examined in SEI (Secondary Electron Imaging) mode using a Hitachi S4000 FE-SEM.

### 2.9. Rotational Melt Rheology

Temperature and frequency sweeps were obtained by an AR G2 rheometer (TA instruments, New Castle, DE, USA) equipped with a 25 mm parallel plate geometry inside of an Environmental Test Chamber (ETC) oven. The gap was 1mm. A strain sweep was conducted to determine the linear viscoelasticity region (<0.05). A temperature sweep from 90 °C to 200 °C was conducted at 1rad/s and a heating rate of 2 °C/min. Frequency sweeps from 0.1 to 600 rad/s were obtained at different temperatures and shifted into a master curve based on the principal of time–temperature superposition (TTS). In addition, Carreau-Yasuda model [35] was used to fit the master curve obtained.

### 2.10. Melt Flow Index Using Extrusion Plastometer

A Tinius Olsen MP 600 Extrusion Plastometer with single barrel was used. The polymer was equilibrated for 7 min at the analysis temperature. At the end of preheat period, the weight platform was lowered to allow the extrudate to fully purge from the cylinder, making note of the time.

### 2.11. Steady-State Viscosity Using Capillary Rheometer

Capillary rheometry was performed using a Rosand RH2000 Capillary Rheometer (Netzsch Group, Germany). A $1 \times 16 \times 180 \times 15$ die (1 mm die diameter $\times$ 16 mm length $\times$ 180 degree angle $\times$ 15 mm bore diameter) was inserted into the RH2000 left bore. A 3000 psi maximum-pressure transducer was used for the HPC GXF, and a 30,000 psi maximum-pressure transducer was used for the HPMC HME 4M, during steady shear rate viscosity measurements. The temperature for measurement was set at 160 °C. Polymers were evaluated at different shear rates ranging from $1 \ \mathrm{s}^{-1}$ up to $180 \ \mathrm{s}^{-1}$ to examine the influence of shear rate on viscosity.

## 3. Results and Discussion

### 3.1. Processing Conditions for Drug–Polymer Blends

Using the minimum quantity of rate-controlling polymers in formulations, where the controlled release of a highly soluble, high-dose active is desired, is a major challenge for formulators. Twin-screw melt granulation can enable a formulator to use minimal amounts of these polymers to obtain controlled- or extended-release formulations of highly soluble actives. The concentration of a polymer in a melt granulation may vary from 10–30% *w/w* depending on the desired drug release pattern. Some authors showed that the successful melt granulation of metformin HCl can be carried out even with a low concentration of polymers (<10%) [27]. These reports, however, targeted the immediate release of metformin HCl. From our experience, a polymer concentration minimum of

15–25% $w/w$ is required to obtain a controlled drug release with high-molecular-weight HPC and HPMC. Therefore, blends of 75% $w/w$ metformin HCl and 25% $w/w$ polymer were prepared for melt granulation.

Different process variables can impact the final product characteristics in a melt granulation process. These process variables include screw speed, powder feed rate, screw configuration, and temperature profile. The granule properties may be impacted by high or low shear in the extruder barrel. The shear in the extruder barrel (at a specific extrusion temperature) can be varied by changing the screw speed and/or powder feed rate. An increase in feed rate and screw speed may increase the mechanical shear inside the extruder barrel to improve binding and agglomeration, and thus increase the granulation tendency of the polymer.

Care was taken to avoid the possibility that the exposure to high shear provided by screw configuration and feed rate could lead to a complete or partial melting of a drug at relatively high temperature. In this case, because the melted drug may also contribute to granulation, any melting of a drug will make it difficult to distinguish between the effects of the melted drug and polymer. To avoid such a situation, only API (metformin hydrochloride) was first passed through the barrel using specific temperature profiles, feed rates, screw configurations, and screw speeds planned for subsequent twin-screw melt granulation experiments. Screw shafts were then pulled out of the barrel. Both screw shafts, as well as the extruder barrel, were examined for any charring or the presence of molten, viscous material which could indicate the possible melting of the drug. If charring or the presence of molten material was observed, those process conditions were eliminated from any future experiments. Usually, the extrusion temperature is kept 30–40 °C below the melting point of the drug. Our initial temperature profile, based on preliminary experiments with HPMC HME 4M and metformin HCl, showed that the drug partially melted at the temperature of 180 °C. Therefore, the extrusion temperature of 180 °C was removed from the experimental design and a maximum extrusion temperature of 160 °C was used for all experiments. A feed rate of 20 g/min and screw speed of 100 rpm was kept constant for all experiments.

Two different screw configurations were used for this study, as shown in Figure 1. Screw configuration#2 provided moderate shear to the powder, with mixing elements rotating at 60°. Screw configuration#1, with mixing elements rotating at 90°, provided higher shear to the material. Blends were melt granulated at four processing temperatures of 160, 140, 120 and 100 °C using a powder feed rate of 20 g/min and screw speed of 100 rpm. The extrusion temperature and screw configuration were changed and the effect of these two process variables on granule properties was studied. Different temperature profiles used for this study are shown in Table 2.

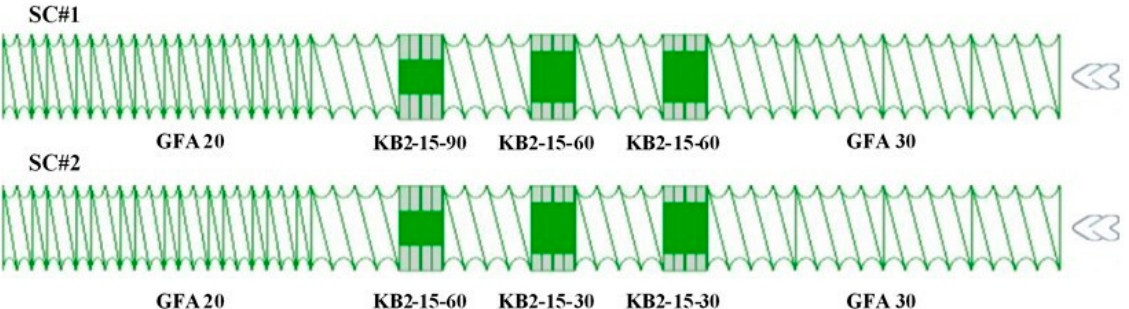

**Figure 1.** Screw configurations used for twin-screw melt granulation. GFA 30 (conveying) and GFA 20 (conveying) represent conveying screw elements with pitch 30 and 20 mm, respectively. KB2-15-30, KB2-15-60 and KB2-15-90 represent blocks of kneading elements with screw elements rotating at an offset angle of 30, 60 and 90°, respectively. Screw configuration#1 (SC#1) with kneading elements rotating at 60 and 90° provided higher shear to the material than screw configuration#2 with kneading elements rotating at 30 and 60°.

**Table 2.** Process parameters used for melt granulation of metformin HCl (75% *w/w*) and polymer (25% *w/w*).

| Feed Rate (g/min) | Screw Speed (rpm) | Zone 1 (°C) | Zone 2 (°C) | Zone 3 (°C) | Zone 4 (°C) | Zone 5 (°C) | Zone 6 (°C) | Zone 7 (°C) | Zone 8 (°C) |
|---|---|---|---|---|---|---|---|---|---|
| 20 | 100 | 20 | 60 | 100 | 100 | 100 | 100 | 100 | 100 |
| | | 20 | 60 | 100 | 120 | 120 | 120 | 120 | 120 |
| | | 20 | 60 | 100 | 140 | 140 | 140 | 140 | 140 |
| | | 20 | 60 | 100 | 160 | 160 | 160 | 160 | 160 |

When extruding blends of metformin HCl and HPC GXF with screw configuration SC#1, a higher extruder torque was observed, ranging from 8–12%. Higher extrusion temperatures of 160 and 140 °C showed a higher operating torque than the lower temperatures 120 and 100 °C. In the case of HPMC HME 4M, a torque of 4–6% was only observed when extruding at 160 °C with screw configuration SC#1. Extrusion at 100, 120 and 140 °C showed minimum extruder torque (≤3%) during the process. When the screw configuration was changed to a lower shear screw configuration SC#2, the torque remained around 8–12% in the case of HPC GXF, whereas melt granulation with HPMC HME 4M showed a minimum extruder torque at all temperatures. Extruder torque is an indication of the force the motor exerts to move the screws to push the material forward. A higher extruder torque in melt granulation sometimes indicates a better agglomeration or compaction of material into dense granules.

*3.2. Granule Properties*

In twin-screw melt granulation, process conditions, as well as the melt viscosity of the binder, can impact the granule properties. Using screw configuration SC#1 for the extrusion of HPC GXF, granule shape and appearance were affected when the processing temperature was varied from 100 to 160 °C, as shown in Figure 2a. At a lower processing temperature of 100 °C, the granules were hard agglomerates (centimeter), but smaller in size. As the temperature was increased from 100 to 120 °C, the granule morphology changed to more ribbon-like granules. At higher temperatures of 140 and 160 °C, the granules produced were more consistently continuous and ribbon-shaped. The granule morphology indicated that, at higher temperatures, HPC had a lower melt viscosity and increased binding capacity to bind and agglomerate metformin HCl to produce ribbon-shaped granules. It was apparent that these granules or extrudates were to be milled to produce smaller particles that could provide an acceptable flow for tableting. When the screw configuration was changed to a low shear SC#1, no significant change in granule morphology was observed. Ribbon-like extrudates were obtained at high temperatures of 140 and 160 °C and hard agglomerates were obtained at 100 and 120 °C.

In the case of HPMC HME 4M, when the processing temperature was varied from 100 °C to 160 °C, HPMC HME 4M granule quality showed a significant variability, as shown in Figure 2b. For HPMC HME 4M, granules were only obtained at 160 °C. At lower temperatures, the polymer did not fully melt, yielding a fine powder and granules of an irregular nature. Nonetheless, it was anticipated that granules obtained with HPMC HME 4M may still provide adequate tablet strength. When the screw configuration was changed to a lower shear, SC#2, fine particles were obtained at all temperatures. There was no change in the appearance of the melt-granulated particles in the feed blend. This showed that HPMC HME 4M required a higher shear in the extruder to melt, and a low shear screw configuration was inadequate for providing shear to the polymer for melting and granulation.

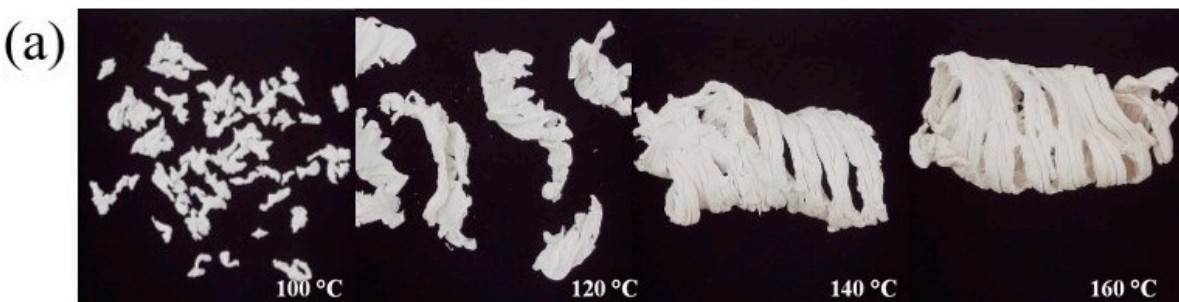

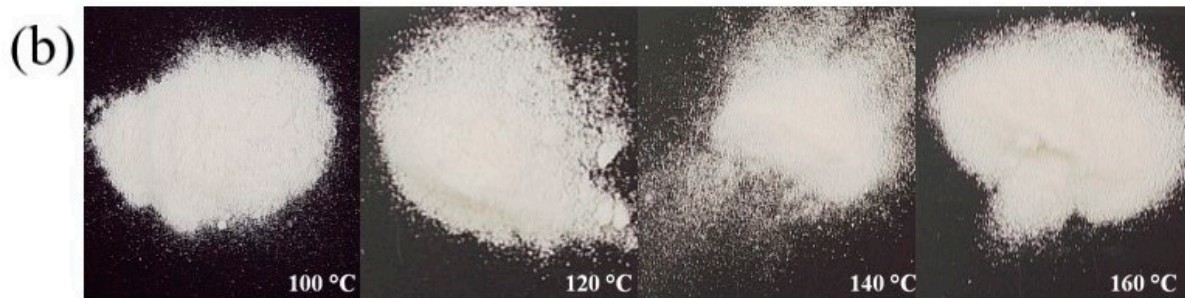

**Figure 2.** (**a**) Extrudates/agglomerates obtained after extrusion of metformin HCl (75%) and HPC GXF (25%) (**b**) Powdery agglomerates obtained after extrusion of metformin HCl (75%) and HPMC HME 4M (25%) at processing temperatures of 100, 120, 140, and 160 °C. High shear screw configuration SC#1 was used for the process. Agglomerates obtained in the case of HPC GXF were irregular and larger in size at a lower processing temperature of 100 °C. At processing temperatures of 120, 140, and 160 °C, the agglomerates looked like ribbon-shaped extrudates, thereby indicating excellent binding capacity. With HPMC HME 4M, fine granules were obtained at 100, 120, 140 °C. At processing temperature of 160 °C, the granules were slightly larger with considerably lower proportion of fine particles.

### 3.3. Differential Scanning Calorimetry (DSC)

Samples obtained by melt granulation of metformin HCl (75% *w/w*) and polymer (25% *w/w*) were analyzed for drug crystallinity using DSC. In all cases, a distinct melting peak in the temperature range of 219–222 °C was observed, indicating that the metformin HCl remained in a crystalline form, post melt granulation with HPMC HME 4M or HPC GXF.

### 3.4. Milling and Sieve Analysis

Granules obtained from melt granulation of metformin HCl with HPC GXF and HPMC HME 4M were milled using a co-mill. After milling samples were collected using screw configuration SC#1 in melt granulation, the ribbon-shaped extrudates were reduced to a powder. Particles obtained after milling of 100 and 120 °C granules showed a higher number of particles, less than <125 μm compared to 140 and 160 °C. This was due to the higher binding capacity of HPC GXF at a higher temperature. When melt-granulated samples, obtained with lower shear screw configuration SC#2, were milled, the particle size distribution remained comparable to that obtained with SC#1. Because these agglomerates are extremely hard and dense, some of these hard agglomerates remained in the mill and did not pass through the screen. The overall yield after milling was 85–90%.

In the case of HPMC HME 4M, milled granules obtained at 100, 120 and 140 °C with screw configuration SC#1 simply passed through the screen during milling. When this powder was analyzed using sieve analysis, most of the particles (>98%) in these samples were <125 μm. Granules obtained at 160 °C also passed through the screen without any potential milling. When these granules were analyzed using sieve analysis for particle size distribution, these powders had a lower number of fines (<125 μm). The particle size

distribution of milled granules obtained at 140 and 160 °C with HPC GXF and HPMC HME 4M is shown in Table 3. When melt-granulated samples of metformin HCl and HPMC HME 4M, obtained using a low shear screw configuration, were milled, all samples simply passed through the screen. When these milled granules were analyzed by sieve analysis, in all cases more than 98% of particles were less than 125 µm. Therefore, only when using a high shear screw configuration SC#1 and high temperature of 160 °C, could a lesser proportion of fines (<125 µm) be obtained in the case of HPMC HME 4M. The powder yield after milling in all cases was close to 100%, as there were no hard agglomerates present in this case, and thus no retention on the screen.

**Table 3.** Particle size distribution of granules obtained after milling melt-granulated samples of metformin HCl and polymeric binder.

| Binder | Screw Configuration | Extrusion Temperature (°C) | <125 µm (%) | 125–300 µm (%) | 300–450 µm (%) | 450–600 µm (%) |
|---|---|---|---|---|---|---|
| HPC GXF | SC#1 | 140 | 83.2 | 16.0 | 0.7 | 0.1 |
|  | SC#1 | 160 | 80.5 | 18.2 | 1.1 | 0.2 |
| HPMC HME 4M | SC#1 | 140 | 98.9 | 1.1 | - | - |
|  | SC#1 | 160 | 91.2 | 8.7 | 0.1 | - |

*3.5. Tablet Compaction*

Tablets were compacted using the milled granules at compaction forces of 15 and 30 kN. In the case of HPC GXF, tablets with acceptable hardness were obtained in all cases irrespective of the screw configuration or processing temperature. This showed that HPC GXF has a high binding capacity, even at low temperatures of 100 and 120 °C and with a minimum shear contributed from the screw configuration SC#2. HPMC HME 4M tablets obtained with extruded granules at 100, 120 and 140 °C were capped on compression. Only the HPMC HME 4M extrudate processed at 160 °C, with the high shear screw configuration SC#1 yielding coherent, strong tablets suitable for dissolution testing. No tablets were obtained with extrudates processed at any temperature with the low shear screw configuration SC#2. This indicated that HPMC HME 4M required a higher shear and temperature to fully melt and provide binding capacity. In contrast, HPC GXF-containing formulations were readily extrudable, yielding a consistent, molten extrudate over the entire processing temperature range of 100 to 160 °C. All the formulations that yielded tablets are shown in Table 4.

**Table 4.** Tablet compaction of milled samples. Tablets with acceptable hardness were obtained in the case of HPC GXF at all temperatures with screw configurations SC#1 and SC#2. In the case of HPMC HME 4M, intact tablets were obtained when the extrusion temperature was 160 °C and with the high shear screw configuration SC#1.

| Binder | Screw Configuration | Compaction Force (kN) | Processing Temperature (°C) | | | |
|---|---|---|---|---|---|---|
|  |  |  | 160 | 140 | 120 | 100 |
| HPC GXF | SC#1 | 15 | + | + | + | + |
|  |  | 30 | + | + | + | + |
|  | SC#2 | 15 | + | + | + | + |
|  |  | 30 | + | + | + | + |
| HPMC HME 4M | SC#1 | 15 | + | - | - | - |
|  |  | 30 | + | - | - | - |
|  | SC#2 | 15 | - | - | - | - |
|  |  | 30 | - | - | - | - |

"+" indicates that an intact tablet was obtained. "-" indicates that all tablets capped during tableting and no intact tablet could be collected.

### 3.6. Dissolution and SEM

Both HPMC HME 4M and HPC GXF have a similar molecular weight and viscosity; they are used for controlled-release tablet formulations. Tablets obtained from melt-granulated samples of metformin HCl with HPMC HME 4M and HPC GXF compressed at a compaction force of 15 kN were selected for dissolution studies.

These tablets yielded similar dissolution profiles, regardless of the process temperature when compared at similar compaction forces. HPC GXF tablets compressed at 30 kN showed a further decrease in dissolution profiles which could be attributed to a decreased porosity. Dissolution profiles for tablets with melt-granulated samples of metformin HCl and HPMC HME 4M or HPC GXF were similar in the late time phase but HPMC HME 4M-based formulations had a higher burst effect compared to HPC GXF (69% in the case of HPMC HME 4M as compared to 44% $w/w$ for HPC GXF at one hour) as shown in Figure 3.

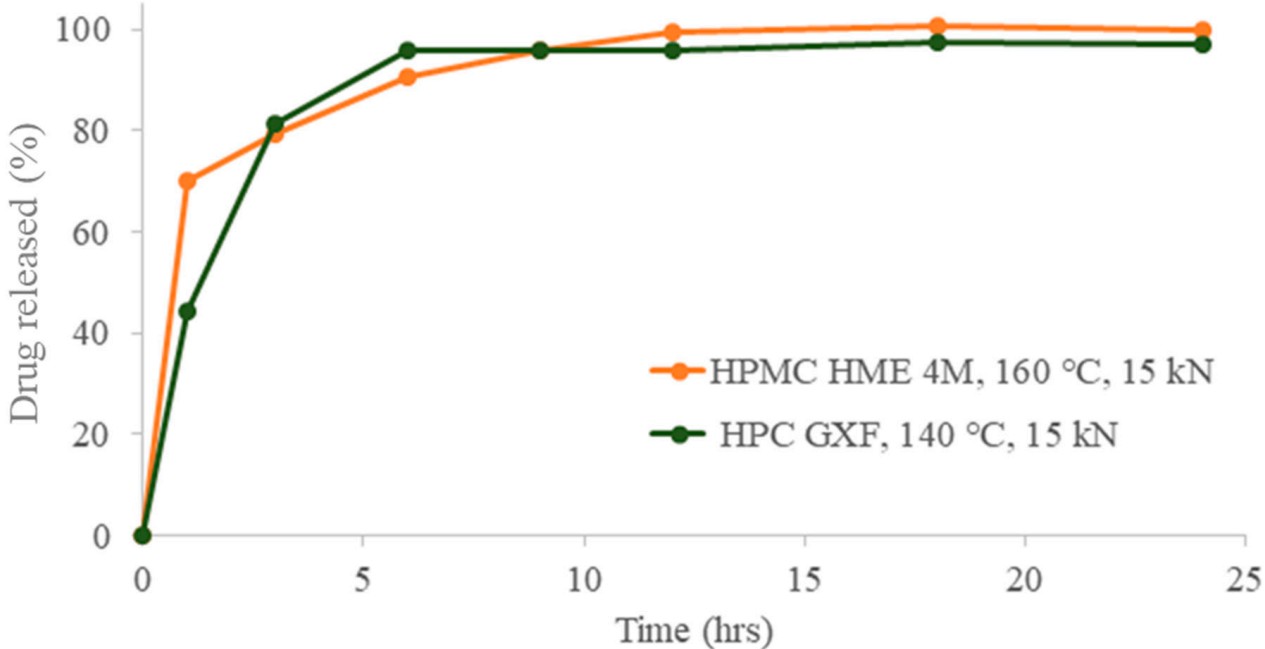

**Figure 3.** Dissolution profiles for tablets obtained by compaction of melt-granulated samples of metformin HCl (75% $w/w$)–HPC GXF (25% $w/w$) and metformin HCl (75% $w/w$)–HPMC HME 4M (25% $w/w$). Tablets with metformin HCl–HPMC HME 4M granules showed burst release of the drug in 1 h (~69%). Tablets with HPC GXF showed comparatively slower release (~44%) in 1 h. The burst release in the case of HPMC HME 4M tablets was due to the formation of large pores which facilitated faster diffusion of the drug out of the matrix.

Since the molecular weights of HPC GXF and HPMC HME 4M are comparable, both the polymers should provide similar swelling and drug release profiles. To investigate the burst release of the drug seen in the case of HPMC HME 4M, the SEM of the edge of the tablet was performed after one hour of the dissolution study. SEM micrographs showed the presence of large pores in the tablet in the case of HPMC HME 4M (shown in Figure 4), which explained the faster release of the drug through the matrix compared to HPC GXF. Tablets with HPC GXF showed very small pores in the matrix.

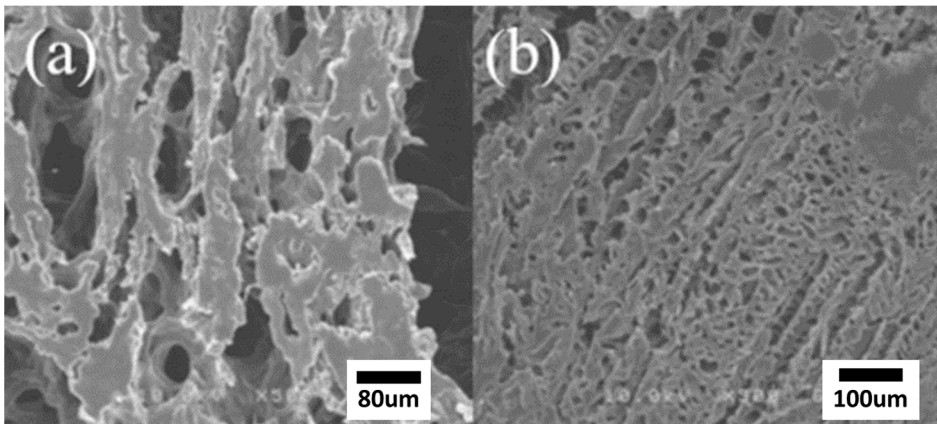

**Figure 4.** SEM of the edge of tablets removed from dissolution media (pH 6.8 phosphate buffer) after 1 h (**a**) Tablet with HPMC HME 4M (**b**) Tablet with HPC GXF. Tablet obtained after melt granulation of metformin HCl (75% *w/w*) and HPMC HME 4M (25% *w/w*) showed large pores which allowed faster release of drug from polymer matrix.

Rotational melt rheology: To provide efficient binding and granule properties, a polymer is required to regulate melt viscosity, so that it can deform and bind the particles together to provide granules. The rheological analysis of the polymer can guide a formulator on minimum processing temperature required for melt granulation process. The granulation capacity of the polymer may depend on its melt viscosity; therefore, a minimum processing temperature is required for a polymer to have an adequate low viscosity (or plasticity), so that it can deform itself and glue the drug particles to form the granules. In general practice, a parallel plate rheological study provides a basic idea on the selection of the minimum extrusion temperature required for the polymer. Glass transition temperature measured by differential scanning calorimeter (DSC) is not a sufficient indicator for guiding a formulator to choose the minimum extrusion temperature. The minimum extrusion temperature required by the polymer is often many degrees higher than its glass transition temperature measured by DSC.

Because the extrusion process is more relevant to the polymer's rheological properties, the minimum extrusion temperature could be indicated by the damping factor and tan δ value from a rheological temperature sweep. The minimum extrusion temperature is often higher than the polymer's glass transition temperature, beyond which the tan δ starts increasing as a function of the temperature, indicating that the polymer chains overcome the intra- and inter-chain entanglements and start the plastic deformations. A temperature ramp from 90 °C to 200 °C is applied to investigate the effect of the temperature on the rheological behaviors of HPMC and HPC. The glass transition temperatures of HPMC HME 4M and HPC GXF are reported at 115 °C [31] and 84 °C [32], respectively; therefore, the temperature ramp tested is directly in the range of interest. The complex viscosity and tan δ as a function of the temperature are depicted in Figure 5. As expected, both HPMC HME 4M and HPC GXF show decreases in the complex viscosity with the increase in the temperature. Specifically, the two stages' decrease in complex viscosity is observed for HPMC HME 4M because the low temperature end is below its glass transition temperature. The complex viscosity decreases from its glass state, which is above $10^6$ Pa ·s, to a glass-rubbery transition state, which is in between $10^4$ and $10^5$ Pa·s, eventually reaching its plastic state, which is below $10^4$ Pa·s. In contrast, the continuous decrease in the complex viscosity from $10^6$ to $10^4$ Pa·s is monitored for HPC GXF because of its relatively lower glass transition temperature. Although it was reported that the complex viscosity should optimally be in the range of $10^3$ to $10^4$ Pa·s to enable melt extrusion, relying only on the complex viscosity is inadequate for selecting the right operational temperature for melt granulation, because it is not required that the polymer is fluid enough or has a low enough viscosity to dissolve the drug when flowing through the barrel and die in the extruder

during melt granulation [17]. Moreover, the relatively close absolute values of the complex viscosity of both polymers make it very difficult to distinguish which is a better binder for the melt granulation process. However, tan δ seems to be a more reliable indicator for guiding the melt granulation temperature selection rather than the complex viscosity. In Figure 5, tan δ signals successfully capture the glass transition of HPMC HME 4M, reflected by the peak around 119 °C. It is reported that the lower glass transition temperature is sufficient to significantly widen the processing temperature window of this HPMC HME 4M, but the tan δ also indicates that the minimum temperature needed to enable its plastic-dominant deformation is about 159 °C. On the contrary, the tan δ of HPC increases from 90 °C and reaches a semi plateau at 139 °C, indicating the melt granulation of HPC is possible when the temperature is above 90 °C. In summary, HPC should be a better melt granulation binder than HPMC HME 4M, with a broader processing temperature window, while HPMC HME 4M is only processable above 159 °C. The conclusion based on the rheological analysis is consistent with what was observed in melt granulation process.

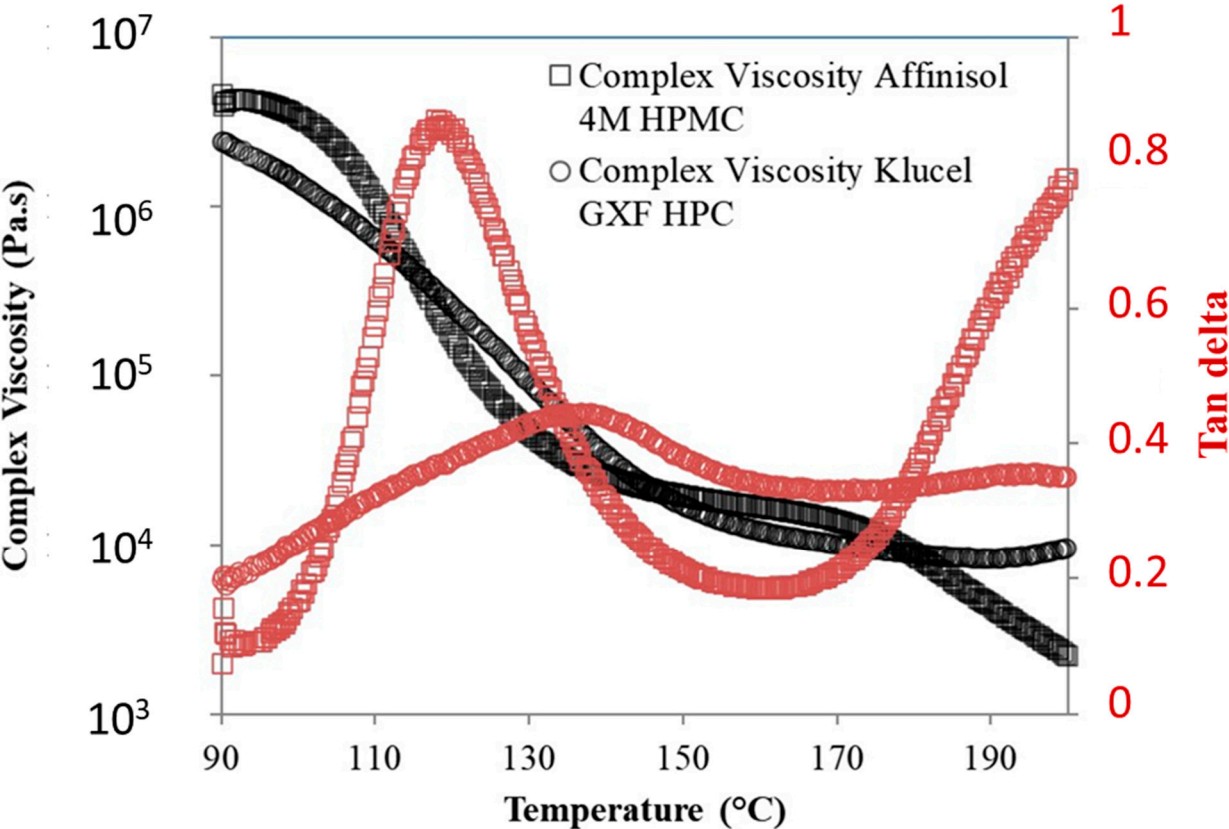

**Figure 5.** Temperature sweep (complex viscosity and Tan delta as a function of temperature) of HPMC HME 4M and HPC GXF using parallel plate rheometer. Both HPMC HME 4M and HPC GXF show decreases in the complex viscosity as the increase in the temperature.

Aside from the processing temperature, the shearing force provided by the extruder contributes to the extrusion and deformation of the polymer. Therefore, a better understanding of the relationship between the shearing force and polymeric deformation is the key to guide robust screw design and the melt granulation process. Considerable efforts were devoted to establishing the structure–processing relationship via a rheological frequency sweep. Measurements of the complex viscosity at various frequencies at a single temperature can show if increasing the shear rate inside the extruder, either by increasing the screw speed or changing the screw configuration, is likely to improve the melt granulation. In this study, frequency sweeps within the linear viscoelastic region are conducted at low strains (<0.05) across an applied frequency range of 0.1 to 600 rad/s at

different temperatures. Subsequently, the frequency sweeps are shifted into one master curve at the reference temperature of 140 °C by means of time-temperature superposition (TTS). Finally, the resulting complex viscosity profile from the master curve was fitted to the Carreau-Yasuda equation, which has a mathematic equation:

$$\eta = \eta_\infty + (\eta_0 - \eta_\infty) \times \left[ 1 + \left( \lambda \dot{\gamma} \right)^\alpha \right]^{\frac{(n-1)}{\alpha}} \qquad (1)$$

where $\eta_0$ and $\eta_\infty$ are the zero shear and infinite shear viscosity, $\lambda$ is the relaxation time, $n$ is power law index and $\alpha$ describes the transition region width. It was also reported that the infinite shear viscosity is very difficult to observe experimentally because the shear rate required for detecting it is very high and outside the normal measurement range. In this study, we could not capture the infinite shear viscosity for both samples. To obtain a more accurate curve fitting, $\eta_\infty$ was set to zero.

Figure 6 displays the complex viscosity–frequency data of HPMC HME 4M and HPC GXF in one master curve at the reference temperature of 140 °C. As shown in the figure, the TTS enables us to cover the rheological property of interest at 140 °C over a relatively broad frequency (time) range (~16 orders), which typically is not feasible when utilizing a single-frequency sweep because of the extremely long experimentation times and increased risk of sample degradation. Complex viscosity is one of the most important rheological properties that describes the processability of a material and can be directly correlated with molecular structure. In this study, both polymers show a typical shear-thinning behavior of polymeric melt, thus their complex viscosities display a strong dependence on the shear rate. Interestingly, both polymers show similar zero-shear viscosity due to their comparable molecular weights. Dependent on the substitution chemistry differences, however, the complex viscosity of HPC displays a stronger sensitivity to shear rate than HPMC does, manifested by a more pronounced shear thinning performance. In addition, other rheological properties, including relaxation time, the width of the transition region and power law index, have also changed correspondingly.

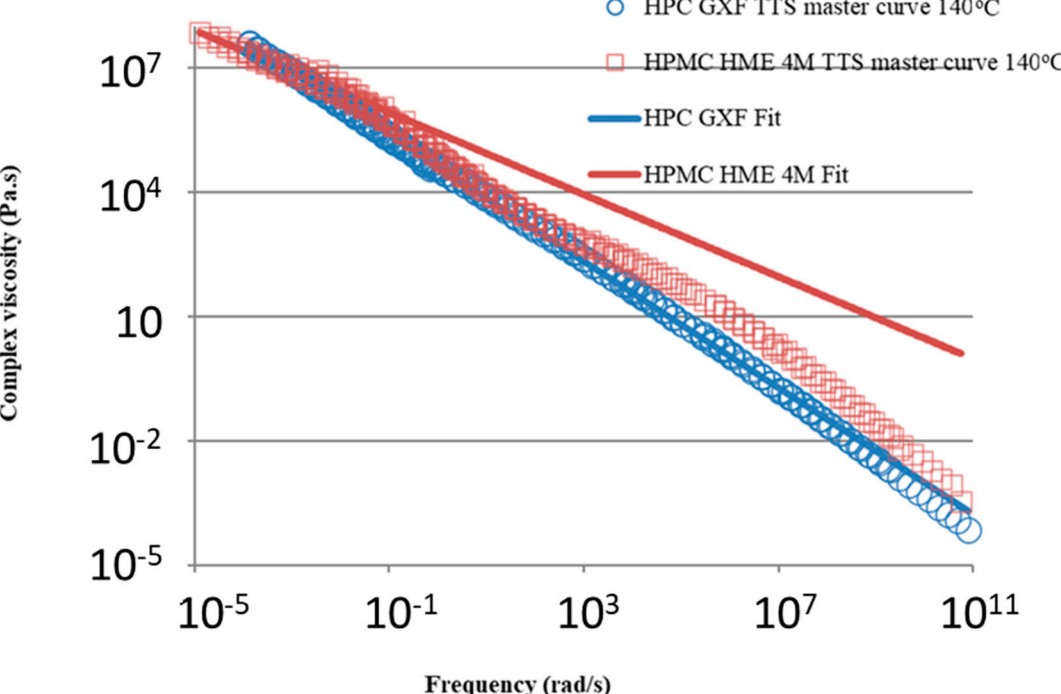

**Figure 6.** Complex viscosity vs. frequency data of HPMC HME 4M and HPC GXF in one master curve at the reference temperature of 140 °C. The modified Carreau-Yasuda model was used to fit the experimental data.

As a result, the master curves exhibited in Figure 6 are in good agreement with the modified Carreau-Yasuda model and the parameters fitting is conducted with sufficient accuracy. All obtained model parameters are summarized in Table 5. In this manner, the rheological properties of HPMC HME 4M and HPC GXF, including the first Newtonian plateau, transition region, and shear thinning region, can be described, and compared quantitatively. Based on the fitting relaxation time, HPC shows one order less relaxation time than HPMC, indicating that it is more plastic and more easily deformed than HPMC under this temperature, making it a better melt granulation binder.

**Table 5.** Model parameters for master curves obtained by fitting complex viscosity vs. frequency curves for HPC GXF and HPMC HME 4M.

| Binder | Carreau-Yasuda Model ($T_r$ = 140 °C) | | | | |
| --- | --- | --- | --- | --- | --- |
| | $\eta_0^*$ (Pa·s) | n | $\alpha$ | $\lambda$ (s) | $r^2$ |
| HPMC HME 4M | $1.5722 \times 10^8$ | 0.50855 | 28.6 | 404352 | 0.9837 |
| HPC GXF | $1.3485 \times 10^8$ | 0.24315 | 16.6 | 41061 | 0.9921 |

*3.7. Melt Flow Index Using Extrusion Plastometer*

In addition to a high temperature, a polymeric material is also under pressure during the extrusion process. While the time–temperature superposition principle is beneficial for understanding polymer rheology, the pressure dependence of rheology is frequently ignored. This is in part because it is more challenging to address with standard instruments. For many rheological tests, relaxation times are collected from measurements on open systems where it is not feasible to pressurize the sample. Extrusion plastometers and capillary rheometers use pressure applied via gravimetric weights to study polymer rheology and provide a better indication of melt flow under pressure; therefore, this process could provide an explanation into the different granulation capacity of HPMC HME 4M and HPC GXF in relation to their melt viscosity.

Melt flow index is defined as mass of polymer (in grams) collected in ten minutes through a capillary of specific length and diameter. Pressure is applied via prescribed gravimetric weights. Figure 7 shows a schematic of the extrusion plastometer used to measure the melt flow index of a polymer. ASTM D1238method is used to calculate melt flow index, where a weight of 10 kg is applied to the polymer, and the time taken by the polymer to pass through the capillary of 2.095 mm is noted. HMPC HME 4M (4.50 g) was added to the capillary maintained at a temperature of 160 °C, and the time required by the polymer to completely exit the capillary was noted. After one hour, only 0.5 g of extruded polymer was collected. It was realized that keeping the polymer at such a high temperature for more than one hour could cause potential degradation and provide false data. Therefore, the standard ASTM method was modified to accommodate a higher gravimetric weight of 20 kgs to push the HPMC HME 4M through the orifice more quickly, and a capillary with a wider orifice diameter of 3.98 mm was installed (Figure 7). The flow of the molted polymer was now measured using this modified ASTM method. The melt flow index of HPMC HME 4M and HPC GXF using the modified ASTM method is listed in Table 6. HPMC HME 4M required an hour to completely exit the capillary, whereas the extruded HPC GXF completely came out of the capillary in 5 s. HPC GXF, due to its significantly higher melt flow index than HPMC HME 4M, possesses a higher thermoplasticity; therefore, it provides a better binding capacity under a high temperature and pressure in the extruder barrel.

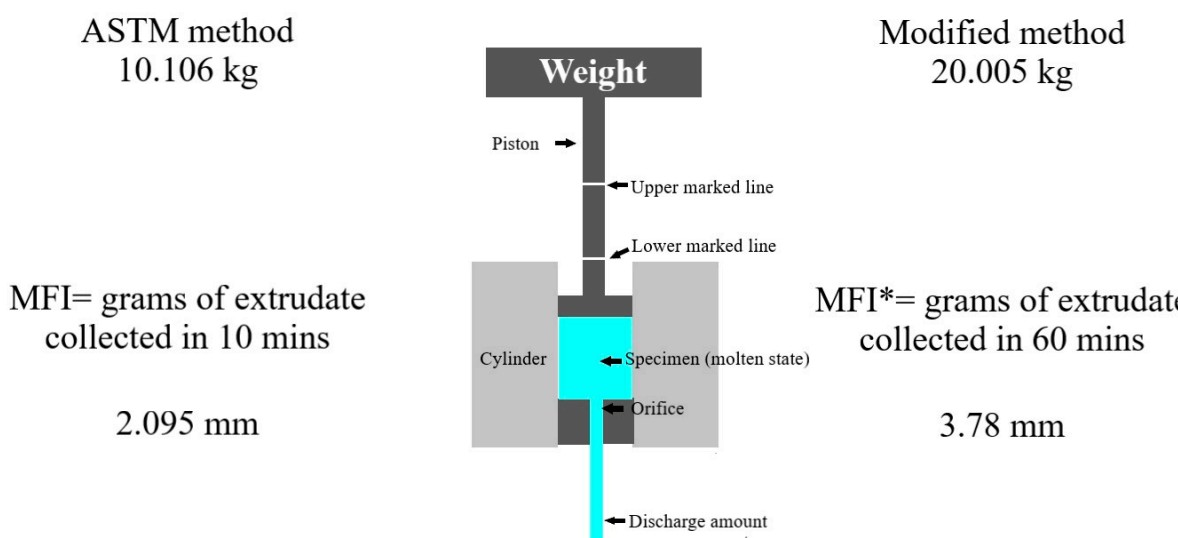

**Figure 7.** Melt flow index of HPMC HME 4M and HPC GXF using ASTM and modified ASTM method at 160 °C. Melt flow indices obtained from the modified method were used to compare ease of extrudability and plasticity of HPMC HME 4M and HPC GXF.

**Table 6.** Melt flow index of HPC GXF and HPMC HME 4M at 160 °C using modified ASTM method.

| Polymer | Initial Sample Weight (g) | Extrudate Collected (g) | Time (s) | MFI* (g/min) |
|---|---|---|---|---|
| HPC GXF | 4.50 | 3.70 | 5 | 44.39 |
| HPMC HME 4M | 4.50 | 3.68 | 3600 | 0.06 |

MFI* is melt flow index obtained using modified ASTM method. ASTM melt flow index method uses 10 kg of weight to push material through a 2.095 mm die and calculates the melt flow index at a specific temperature as grams of extrudate collected in 10 min using this set up. The modified method used for this study utilized a 20 kg weight and 3.78 mm die, calculating the melt flow index as grams of extrudate collected in 60 min.

### 3.8. Steady-State Viscosity

The steady shear-rate viscosity can indicate the differences between polymeric materials under stress. The steady shear-rate viscosity of HPMC HME 4M and HPC GXF was analyzed using a capillary rheometer. Figure 8 shows the steady shear viscosity of the HPMC HME 4M compared to HPC GXF at a temperature of 160 °C, as measured using a capillary rheometer. Both polymers are shear-thinning, indicating that the melt granulation process can be improved in certain cases by changing either the screw rotation speed, the screw diameter, or the channel depth to improve the shear rate inside the extruder. However, the figure also shows that the viscosity is an order of magnitude higher for HPMC HME 4M than for the HPC GXF polymer at all shear rates. This explains why melt granulation is significantly easier with HPC GXF. It is difficult for a polymer to interact with the drug if it does not flow well under shear stress, as is the case with HPMC HME 4M, while the drug interaction with HPC GXF is much easier due to the polymer's lower viscosity.

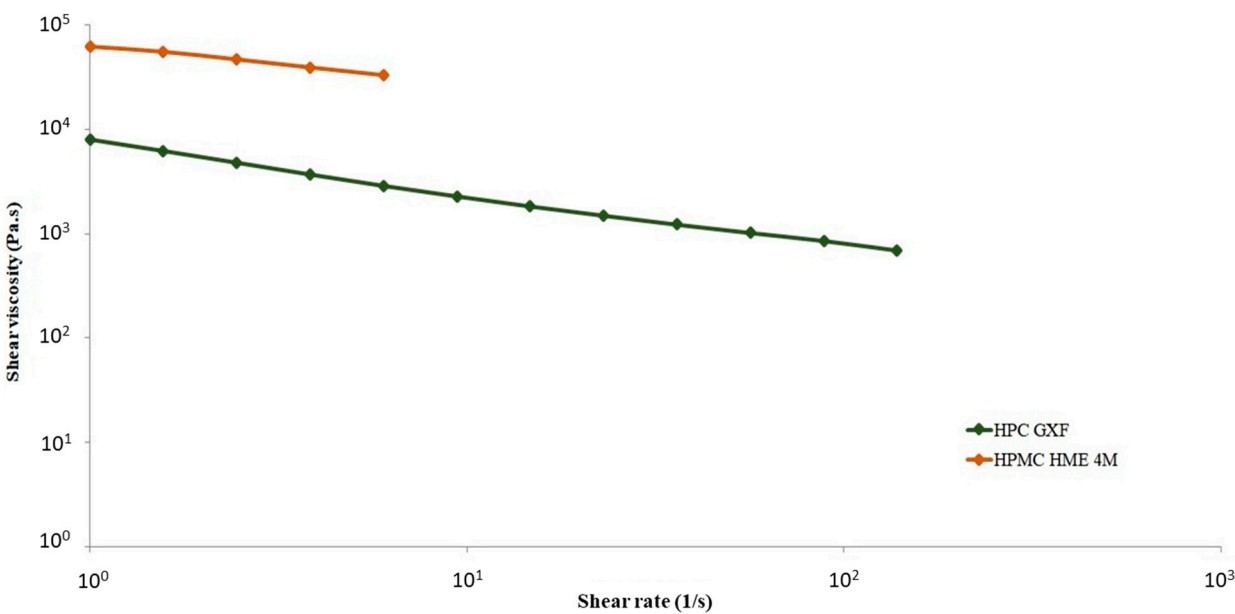

**Figure 8.** Steady-shear viscosity of the HPMC HME 4M and HPC GXF at a temperature of 160 °C as measured using a capillary rheometer. Both polymers are shear thinning, indicating that the melt granulation process can be improved by increasing the shear rate inside the extruder.

## 4. Concluding Remarks

Due to the advances in pharmaceutical manufacturing, especially after the advent of continuous manufacturing, the twin-screw extruder gained popularity for continuous granulation. Melt granulation using a twin-screw extruder emerged as a robust technology for the development of modified-release oral formulations for high-dose and highly water-soluble actives. Although natural polymers such as guar gum [36,37]; gum acacia [38], etc.; and waxy binders such as polyethylene glycol [39] were conventionally used for controlled-release formulations, polymeric binders provide a better control over granule properties in twin-screw melt granulation [10]. The pharmaceutical applications of polymeric materials encompass many diverse areas, and controlling or modifying the release of drug from a dosage form has been a critical application for decades. The controlled delivery of a drug refers specifically to the precise control of the rate at which a particular drug dosage is released from a delivery system (ideally in a constant or near-constant manner over a prolonged period of time) without the need for frequent, repeated administration, either orally or parenterally.

High-molecular-weight HPC and HPMC were used extensively to obtain the controlled or modified release of highly water-soluble actives. Both HPMC HME 4M and HPC GXF can be used to obtain metformin granules for extended-release tablets. HPC GXF has a wider processing window allowing for a low temperature extrusion from 100 °C to 160 °C and more robust processes and formulations. HPMC HME 4M, on the other hand, requires a minimum extrusion temperature of 160 °C, to provide adequate binding capacity and granulation. Tablets produced from granules obtained by extrusion of metformin HCl and HPMC HME 4M showed higher porosity, and thus the initial burst release of the drug. Granules produced with HPC GXF were denser and therefore, required more time for water to penetrate and the drug to diffuse out.

A rheological analysis revealed that temperature and frequency profiles for HPMC HME 4M and HPC GXF were similar; therefore, both of the polymers had similar melt viscosity at zero shear. The temperature and frequency sweep using the parallel plate rheometer indicated that both HPMC HME 4M and HPC GXF should provide similar granulation and binding capacities, since both had a similar melt viscosity at the extrusion temperature. Steady-state viscosity using capillary rheometer showed that there was a difference in the steady shear viscosity of the two polymers, and thus the difference in

melt granulation behavior. The excellent binding capacity of HPC was also noted by Batra et al. [10] and Kittikunakorn et al. [40]. The authors observed that HPC could be extruded at temperatures as low as 130 °C with only 5% of the binder and provided an excellent binding capacity and tensile strength of tablets. The authors concluded that this capability of HPC to be extruded at such low temperatures could be attributed to its thermoplasticity.

Significant scientific efforts are required in the field of twin-screw melt granulation to understand the granulation behavior with different polymeric binders. As the field of continuous granulation expands to more and more commercial products, such scientific reports will enhance the knowledge of formulators and provide guidance in designing the melt granulation processes for new products. Given its multiple advantages, melt granulation technology holds a significant potential in the future.

**Author Contributions:** Conceptualization, A.B., F.Y. and T.D.; methodology, F.Y.; investigation, A.B. and F.Y.; Data curation, A.S., C.U., E.W.O., N.C. and K.L.; Investigation, Y.B.; writing—original draft preparation, A.B., M.K. and F.Y.; supervision, T.D. All authors have read and agreed to the published version of the manuscript.

**Funding:** This research received no external funding.

**Institutional Review Board Statement:** Not applicable.

**Informed Consent Statement:** Not applicable.

**Data Availability Statement:** The data presented in this study are available on request from the corresponding author.

**Conflicts of Interest:** The authors have no competing interests.

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
