# Peer review of "Comparison of Hydroxypropylcellulose and Hot-Melt Extrudable Hypromellose in Twin-Screw Melt Granulation of Metformin Hydrochloride: Effect of Rheological Properties of Polymer on Melt Granulation and Granule Properties"

_2673-6209, doi:10.3390/macromol2010001_

Round 1

Reviewer 1 Report

The authors of the work investigate the technological properties of compositions based on hypromellose and hydroxypropyl cellulose during melt granulation. Particular attention is paid to the study of the rheological properties of polymer melts.

There are several comments on the work, I hope that the authors will take them into account.

I recommend removing the abbreviations HPC and HPMC from the title of the article. Leave explanations only in the text of the article. If possible, shorten the title.

Line 24 Replace "parallel plate rheology" with "Rotational rheometry"

Line 85. "…even when used at lower concentrations of 10-30%" - add link

Line 112 Number all headings and subheadings

Line 160 Heating rate?

Line 206. Replace with «Rotational Rheometry». This section should describe in detail the characteristics of the measuring unit and the parameters of the experiment.

What was the gap between the parallel plates? Have the plates been profiled?

It is necessary to describe in detail the parameters of oscillatory measurements - frequency, strain amplitude, etc. Has the range of linear viscoelasticity been determined?

What is the heating rate during temperature testing? Is the heating linear or stepped?

Line 233 "Some authors ...". You need to add a link.

Line 238. "25% w / w" - of what?

Lines 265-268, 278-280. "Two different screw configurations were used for this study as shown in Figure 1. Screw configuration # 1 provided moderate shear to the powder with mixing elements rotating at 60°. Screw configuration # 2 with mixing elements rotating at 90° provided higher shear to the material. "

  "Screw configuration # 1 (SC # 1) with kneading elements rotating at 60 and 90° provided higher shear to the material than screw configuration # 2 with kneading elements rotating at 30 and 60°."

These are two opposite statements.

Line 276. GFA - decryption required

Line 298. "smaller in size" - Better to indicate the size range in conventional units

Line 320. Figure 2. I recommend to indicate the scale in the figure

Line 327. I recommend the authors to present the DSC results graphically (it is possible in supplementary materials so as not to overload the article).

Figure 3. "Drug dissolved" is better to replace with "drug release"

Line 408. Figure 4. You need to add a scale bar.

Line 412 Replace with "Rotational Rheometry"

Line 427. "Tan delta" - it is necessary to clarify, not all readers will understand what this value is.

Line 440. Hereinafter - correct the "dot" sign in the designation of viscosity measurement units ("Pa ∙ s" - the "multiply" sign in the form of a dot must be indicated).

Lines 443-448. Add link.

Line 464. Figure 5. Caption contains only complex viscosity. It needs to be supplemented.

Figures 5, 6 and 8 should be in the same style. Designate the orders as 10 to the appropriate power (101, 102, 103, etc.). In Figures 5 and 6, reduce the number of points on the curves to make the drawing more readable (the "skip point" function will not affect the results). In Figure 8 - enlarge the labels on the axes, by analogy with Figures 5 and 6.

Line 480. How justified is the application of the specified model with the use of complex dynamic viscosity? Does the Cox-Merz rule hold for your systems? I recommend the authors to measure the shear viscosity of the systems using a rotary rheometer, compare the obtained values of viscosity with the values of shear viscosity on a capillary rheometer.

I recommend that the authors pay more attention to the concepts of "complex dynamic viscosity" and "shear viscosity".

Lines 515-517. Remove - "Based on the fitting relaxation time, HPC shows one order less relaxation time than HPMC, indicating it is more plastic and is easier to be deformed than HPMC under this temperature, also making it is a better melt granulation binder."

These results raise questions. A detailed description is needed - how did the authors apply the temperature-time superposition principle? How was the shear factor calculated? What relaxation times did you use?

Lines 534, 541. You must specify the height of the channel (Lines219-220 other information is indicated - A 1x16x180x15 die (1mm die diameter x 16mm length x 180 degree angle x 15mm bore diameter))

Figure 7 - you can delete this figure, it is not informative.

Lines 549-553. Remove - "HPMC HME 4M did not extrude out of the capillary even after one hour. ASTM method was modified to accommodate higher weight to increase pressure on the molted polymer and larger orifice to extrude HPMC HME 4M in shorter time. Melt flow indices obtained from the modified method were used to compare ease of extrudability and plasticity of HPMC HME 4M and HPC GXF. "

Figure 8. Delete - "Both polymers are shear thinning indicating that the melt granulation process can be improved by increasing shear rate inside the extruder." Correct the legend (1/s).

Authors should correct the list of references in accordance with the requirements of the journal.

Link 11???

Author Response

Dear reviewer,

First, we apricate your time and effort on reviewing our manuscript. Also, we want to thank you for your suggestion and correction. Here, we are happy to provide you a point to point reply to resolve all the questions or concerns you may have. Thanks!

The authors of the work investigate the technological properties of compositions based on hypromellose and hydroxypropyl cellulose during melt granulation. Particular attention is paid to the study of the rheological properties of polymer melts.

There are several comments on the work, I hope that the authors will take them into account.

I recommend removing the abbreviations HPC and HPMC from the title of the article. Leave explanations only in the text of the article. If possible, shorten the title.

Thank you for the suggestion, we have removed the abbreviations of HPC and HPMC, but we have done our best to control the length of the title. Thanks!

Line 24 Replace "parallel plate rheology" with "Rotational rheometry"

This is a good suggestion and we have changed the expression in the manuscript. Thanks!

Line 85. "…even when used at lower concentrations of 10-30%" - add link

We have referred to reference 2. Thanks!

Line 112 Number all headings and subheadings

Line 112 does not have headings and subheadings. Thanks!

Line 160 Heating rate?

The heating rate in the extruder is not a setting value, it is an estimation based on the feeding rate, screw speed and barrel temperature. Since we have changed the barrel temperature for different materials, therefore the heating rate would not be a constant for different processing conditions.  

Line 206. Replace with «Rotational Rheometry». This section should describe in detail the characteristics of the measuring unit and the parameters of the experiment.

This is a good suggestion and we have changed the expression in the manuscript. In addition, detailed discussion was covered in the result section. Thanks!

What was the gap between the parallel plates? Have the plates been profiled?

The gap was 1mm and the information was added to the manuscript. The plates were stainless steel with flat surface.

It is necessary to describe in detail the parameters of oscillatory measurements - frequency, strain amplitude, etc. Has the range of linear viscoelasticity been determined?

This is a very good suggestion and we have added all the information in the experimental section.  Also, the region of linear viscoelasticity was pre-determined by a strain sweep, which was <0.05. 

What is the heating rate during temperature testing? Is the heating linear or stepped?

The heating rate was 2°C/min and the information was added to the manuscript. The heating process is a linear one.

Line 233 "Some authors ...". You need to add a link.

The reference link was updated.

Line 238. "25% w / w" - of what?

The information of polymer has been added.

Lines 265-268, 278-280. "Two different screw configurations were used for this study as shown in Figure 1. Screw configuration # 1 provided moderate shear to the powder with mixing elements rotating at 60°. Screw configuration # 2 with mixing elements rotating at 90° provided higher shear to the material. "

  "Screw configuration # 1 (SC # 1) with kneading elements rotating at 60 and 90° provided higher shear to the material than screw configuration # 2 with kneading elements rotating at 30 and 60°."

These are two opposite statements.

Thank the reviewer for the good catch, we had a typo in the first sentence, we have corrected it in the manuscript. Thanks!

Line 276. GFA - decryption required

GFA stands for co-rotating conveying element, intermeshing. The information had been added.

Line 298. "smaller in size" - Better to indicate the size range in conventional units

As shown in the image, the hard agglomerates formed a continuous phase when the temperature was higher than 100°C. In contrast, the agglomerates were in a discontinues morphology approximately in centimeter length.  The information has been added.

Line 320. Figure 2. I recommend indicating the scale in the figure

This is a good suggestion. However, the image was taken right after the melt granulation and the granules had been milled. Therefore, it is not possible for us to add accurate scale to this image now. Thanks!   

Line 327. I recommend the authors to present the DSC results graphically (it is possible in supplementary materials so as not to overload the article).

All the DSC showing similar thermogram because we had processed the melt granulation way below the melting point of metformin HCl. DSC was utilized to verify the crystallinity of metformin HCl and would not provide any additional information, so we do not think it is necessary to add DSC results graphically.

Figure 3. "Drug dissolved" is better to replace with "drug release"

The figure has been updated.

Line 408. Figure 4. You need to add a scale bar.

The figure has been updated.

Line 412 Replace with "Rotational Rheometry"

The expression has been changed.

Line 427. "Tan delta" - it is necessary to clarify, not all readers will understand what this value is.

The information, damping factor had been added in the manuscript.

Line 440. Hereinafter - correct the "dot" sign in the designation of viscosity measurement units ("Pa âˆ™ s" - the "multiply" sign in the form of a dot must be indicated).

Corrections had been made.

Lines 443-448. Add link.

Reference was added.

Line 464. Figure 5. Caption contains only complex viscosity. It needs to be supplemented.

The caption and figure had been updated.

Figures 5, 6 and 8 should be in the same style. Designate the orders as 10 to the appropriate power (101, 102, 103, etc.). In Figures 5 and 6, reduce the number of points on the curves to make the drawing more readable (the "skip point" function will not affect the results). In Figure 8 - enlarge the labels on the axes, by analogy with Figures 5 and 6.

Figure 8 had been replotted to matched figure 5 and 6.

Line 480. How justified is the application of the specified model with the use of complex dynamic viscosity? Does the Cox-Merz rule hold for your systems? I recommend the authors to measure the shear viscosity of the systems using a rotary rheometer, compare the obtained values of viscosity with the values of shear viscosity on a capillary rheometer.

Carreau–Yasuda Model was one of the most used models (such as powder law model), the model was well established and fit our data very well as showing in the manuscript, r2>0.98. we did not check the Cox-Merz rule in particularly, because shearing viscosity was reported at 160°C while complex viscosity was reported at 140°C. However, if you check the complex viscosity reported in temperature sweep at 160°C, the results were consistent with what we have seen in shearing viscosity in two terms: first HPMC showing higher viscosity over HPC; second the viscosity fall into the same 104 to 105 Pa âˆ™ s range. Finally, due to the high melt viscosity of both polymers, it will be very challenging to run a shear viscosity test on a rotary rheometer rather than run it in an oscillatory mode.

I recommend that the authors pay more attention to the concepts of "complex dynamic viscosity" and "shear viscosity".

We had discussion on these two concepts in two separate section. Because these parameters were measured by using different rheometer with different mechanism and geometry. The compassion between these two concepts had been reported by other rheologist before, but it is not the focus on this paper. Thanks!  

Lines 515-517. Remove - "Based on the fitting relaxation time, HPC shows one order less relaxation time than HPMC, indicating it is more plastic and is easier to be deformed than HPMC under this temperature, also making it is a better melt granulation binder."

These results raise questions. A detailed description is needed - how did the authors apply the temperature-time superposition principle? How was the shear factor calculated? What relaxation times did you use?

Temperature-time superposition principle was a well-established theory and had been widely used to characterize the polymer melt’s behavior under shear. Most of rheological data process software, such as the TA TRIOS we have used, has built-in function to do TTS for a given set of data. It can calculate the shift factors automatically. The relaxation time is a fitting variable, which is calculated based on the experimental data. In general,  The shorter the relation time is, the more plastic (easier to deform) the polymer is.   

Lines 534, 541. You must specify the height of the channel (Lines219-220 other information is indicated - A 1x16x180x15 die (1mm die diameter x 16mm length x 180-degree angle x 15mm bore diameter))

The information had been provided in the experimental section and the reviewer got the right explanation.

Figure 7 - you can delete this figure, it is not informative.

This figure showed the details on how to run a melt index test on the polymers. Also, it provided the comparison between the ASTM method and the modified method we used eventually. Thanks!

Lines 549-553. Remove - "HPMC HME 4M did not extrude out of the capillary even after one hour. ASTM method was modified to accommodate higher weight to increase pressure on the molted polymer and larger orifice to extrude HPMC HME 4M in shorter time. Melt flow indices obtained from the modified method were used to compare ease of extrudability and plasticity of HPMC HME 4M and HPC GXF. "

These key findings highlighted here were something important to discriminate HPC from HPMC for melt granulation process. As we mentioned, we successfully used a standard ASTM method to measure the melt flow index of HPC, however, the method was not applicable to HPMC due to the insufficient weight applied. To enable the test on HPMC, we had modified the ASTM method by using a heavier block. In the end, we noticed the significant melt flow difference between HPC and HPMC, indicating HPC has better extrudability and plasticity.     

Figure 8. Delete - "Both polymers are shear thinning indicating that the melt granulation process can be improved by increasing shear rate inside the extruder." Correct the legend (1/s).

The figure has been updated. The conclusion here was important because the rheological performance of these two polymers were consistent with what we had observed in the melt extrusion process, especially that HPMC melt granulation was enabled when we changed the screw configuration from #2 to #1.    

Authors should correct the list of references in accordance with the requirements of the journal.

We had corrected the list of reference, thanks!

Sincerely,

Fengyuan Yang, Ph.D.

Reviewer 2 Report

Summary 

The manuscript entitled “Comparison of hydroxypropylcellulose (HPC) and hot-melt extrudable hypromellose (HPMC) in twin-screw melt granulation of metformin hydrochloride: Effect of rheological properties of polymer on melt granulation and granule properties’’ by Wang et al. shows the processability and dissolution behavior of HPC GXF ((Klucel® GXF) and a recently introduced type of hot-melt extrudable HPMC (Affinisol®) in extended-release metformin hydrochloride formulations using twin-screw melt granulation. The presented article contains both basic experimental and application tests, which are vital to prove the claim with reliable data and in turn pave the way towards the application of this system.

General comments

In general, the work is accurate and well presented as a Macromol article should be, moreover the presented results are certainly of interest to readers of this journal. The article style is correct, but it should be reviewed in a few points. Thus, I believe that the text needs some technical adjustments to be published. Therefore, I recommend that this manuscript can be published in Macromol after Major revision.

Specific comments

Going into detail on the specific issues, here some comments are reported:

- the article's grammar, punctuation, and style are quite adequate, but the manuscript needs to be slightly proofread.

- it is worth highlighting that cellulose-based polymeric materials can be obtained with other fabrication techniques (e.g. electrospinning: https://doi.org/10.1016/j.tafmec.2018.11.006). highlight it in the Introduction section.  

- all the quantitative data (including the roughness) need to be reported as mean ± STD. The error bars should be indicated in the graphs.

- SEM images are of poor quality and the scale bars should be well visible

Conclusion

The topic of this manuscript falls within the scope of Macromol. I like the concept and material development/characterization proposed in this paper. I believe the article is of sufficient quality and novelty to meet the Macromol publication standard after a major revision.

Author Response

Dear reviewer,

First, we apricate your time and effort on reviewing our manuscript. Also, we want to thank you for your suggestion and correction. Here, we are happy to provide you a point to point reply to resolve all the questions or concerns you may have. Thanks!

Specific comments

Going into detail on the specific issues, here some comments are reported:

- the article's grammar, punctuation, and style are quite adequate, but the manuscript needs to be slightly proofread.

We thank the reviewer for the suggestion, and we competed one more round of proofreading and made necessary corrections and modifications. Thanks!

- it is worth highlighting that cellulose-based polymeric materials can be obtained with other fabrication techniques (e.g. electrospinning: https://doi.org/10.1016/j.tafmec.2018.11.006). highlight it in the Introduction section.  

In this manuscript we focused on the application of cellulose-based polymeric materials (HPC an HPMC) in melt granulation by using twin-screw extrusion. We did not cover the fabrication techniques of cellulose-based polymeric in the introduction at all. It will be difficult for us to add the recommended reference to the introduction section. Thanks!

- all the quantitative data (including the roughness) need to be reported as mean ± STD. The error bars should be indicated in the graphs.

We did not report roughness in this manuscript. Most of the quantitative data in the manuscript such as rheological data was reported by using their averages, but due to the logarithmic scale was used, the error bars were not included. Thanks!

- SEM images are of poor quality and the scale bars should be well visible

The SEM images had been updated with scale bar. Thanks!

Sincerely,

Fengyuan Yang, Ph.D.

Round 2

Reviewer 1 Report

The authors of the work took into account only a part of the comments. A number of questions remained for work.

Line 160. Differential Scanning Calorimetry - Heating Rate? Measurement modes? It is necessary to supplement this section

Line 439. “The complex viscosity and tan δ as a function of the temperature are depicted in Figure 5” At what angular frequency were these data obtained?

Line 472 "Measurements of the complex viscosity at various frequencies at a single temperature..." I recommend showing the corresponding dependences in the article graphically.

Line 492-499 "... their complex viscosities display a strong dependence on the shear rate" The authors of the work replace the concepts of "complex viscosity" and "shear-rate-dependent viscosity"

It is necessary to justify the application of the Carreau – Yasuda Model (equation 1) for the data obtained in the oscillatory measurement mode. What do the authors take for the shear rate? Angular frequency? It is necessary to confirm the legality of such a substitution by links to the corresponding works. Or replace Equation 1 with an equation in which the complex viscosity depends on frequency (for example, as done in doi.org/10.1016/j.rinp.2019.102245). The viscosity frequency dependences obtained on a rotational rheometer should be presented graphically.

Figure 5 is overloaded with graphs. Curves are hard to read. I recommend that the authors prepare two separate drawings (for example, Figure 5-a and Figure 5-b). In one figure, give the data of complex dynamic viscosity, in the other - tangent delta. The number of points needs to be reduced, it is completely incomprehensible where is the "circle" in the figure, and where is the "square". It is necessary to correct the labels for the axes, I wrote about this in the previous review.

Figure 6. Too many points, curves merge. It is necessary to remove some of the points (the "skip point" method) to make the graph more understandable for the reader. Correct numerical designations. The figure caption should only contain a description of the curves. I recommend removing all detailed comments in the text of the article in the appropriate section. It would be nice to put in the "additional materials" a more detailed description of the application of the principle of temperature-time superposition, namely, the initial graphs and the calculated part.

Figure 7. The figure of the scientific article should contain the title and explanation of the graphs/points/curves. All detailed comments/explanations/discussion should be placed in the text of the article, in the appropriate section.

I would like to draw the attention of the authors to the fact that the caption should only contain the name of the figure and a brief explanation of the curves. All conclusions are placed in the main text of the article. I do not recommend overloading pictures.

The list of references has not been corrected.

Author Response

Dear reviewer,

Firstly, we appreciate your time and effort on reviewing our manuscript. Also, we want to thank you for your suggestion and correction. Here, we are happy to provide you a point to point reply to resolve all the questions or concerns you may have. Thanks!

Line 160. Differential Scanning Calorimetry - Heating Rate? Measurement modes? It is necessary to supplement this section

The requested information has been updated in the manuscript. Thanks!

Line 439. “The complex viscosity and tan δ as a function of the temperature are depicted in Figure 5” At what angular frequency were these data obtained?

The information was already covered in the experimental section and the frequency was 1rad/s.

Line 472 "Measurements of the complex viscosity at various frequencies at a single temperature..." I recommend showing the corresponding dependences in the article graphically.

The requested data were busy in a single figure as shown above (individually or TTS overlays) and a master curve based on TTS at a reference temperature can cover all the information interested, which is a more practical and acceptable way to present the frequency sweeps data.

Line 492-499 "... their complex viscosities display a strong dependence on the shear rate" The authors of the work replace the concepts of "complex viscosity" and "shear-rate-dependent viscosity"

It is necessary to justify the application of the Carreau – Yasuda Model (equation 1) for the data obtained in the oscillatory measurement mode. What do the authors take for the shear rate? Angular frequency? It is necessary to confirm the legality of such a substitution by links to the corresponding works. Or replace Equation 1 with an equation in which the complex viscosity depends on frequency (for example, as done in doi.org/10.1016/j.rinp.2019.102245). The viscosity frequency dependences obtained on a rotational rheometer should be presented graphically.

As the data have shown, complex viscosity is shear rate dependent, so we did not replace the concepts. The data shown in the above figure was collected at different temperatures, frequency sweeps were done from 0.1 to 600 rad/s. Based on TTS theory,  and one can shift the curves into a single master curve at any reference temperature that has been covered. In this case, we used140℃. Carreau–Yasuda Model was one of the most used models (such as powder law model), the model was well established and fit our data very well as showing in the manuscript, r2>0.98.

Figure 5 is overloaded with graphs. Curves are hard to read. I recommend that the authors prepare two separate drawings (for example, Figure 5-a and Figure 5-b). In one figure, give the data of complex dynamic viscosity, in the other - tangent delta. The number of points needs to be reduced, it is completely incomprehensible where is the "circle" in the figure, and where is the "square". It is necessary to correct the labels for the axes, I wrote about this in the previous review.

We have updated the figure and made it more reader friendly. Thanks!

Figure 6. Too many points, curves merge. It is necessary to remove some of the points (the "skip point" method) to make the graph more understandable for the reader. Correct numerical designations. The figure caption should only contain a description of the curves. I recommend removing all detailed comments in the text of the article in the appropriate section. It would be nice to put in the "additional materials" a more detailed description of the application of the principle of temperature-time superposition, namely, the initial graphs and the calculated part.

Temperature-time superposition (TTS) principle was a well-established theory and had been widely used to characterize the polymer melt’s behavior under different shear rate. Most of rheological data process software, such as the TA TRIOS we have used, has built-in function to do TTS for a given set of data. It can calculate the shift factors automatically based on the data. We do not think it is necessary to have an educational section on the principle of TTS.

Due to the nature of TTS, omitting data point in a master curve is not allowed, because the master curve was generated by shifting multiple frequency sweep curves and overlay them at a refence temperature. therefore, it is common that the middle part of the master curve has more data points than on its both ends. That is why in our figure, HPMC showed a deviation at the high frequency in the fitting but it still gave a decent r2, because most data point were concentered at low frequency range.

 We have modified the caption. Thanks!   

Figure 7. The figure of the scientific article should contain the title and explanation of the graphs/points/curves. All detailed comments/explanations/discussion should be placed in the text of the article, in the appropriate section.

I would like to draw the attention of the authors to the fact that the caption should only contain the name of the figure and a brief explanation of the curves. All conclusions are placed in the main text of the article. I do not recommend overloading pictures.

Thank the reviewer for the suggestion; we agree that the caption should only cover the key information.  We have made changes in the manuscript. Thanks!

The list of references has not been corrected.

We had created the list of reference by using endnote software and complied the requirement of the journal. Unfortunately, we could not find an endnote style file for Macromol, otherwise we would like to make more changes to fulfil the reviewer’s requests.  thanks!

Sincerely,

Fengyuan Yang, Ph.D.

Reviewer 2 Report

The manuscript can be accepted in the present form.

Author Response

Thanks! 

Round 3

Reviewer 1 Report

The authors of the work did not take into account all the comments.

Author Response

Hi There,

I will submit a rebuttal letter to solve all the concerns. Thanks!

Fengyuan Yang, Ph.D.
